# Bispecific Monoclonal Antibodies in Diffuse Large B-Cell Lymphoma: Dawn of a New Era in Targeted Therapy

**DOI:** 10.3390/cancers17193258

**Published:** 2025-10-08

**Authors:** Mattia Schipani, Matteo Bellia, Carola Sella, Riccardo Dondolin, Mariangela Greco, Abdurraouf Mokhtar Mahmoud, Clara Deambrogi, Riccardo Moia, Gianluca Gaidano, Riccardo Bruna

**Affiliations:** Division of Hematology, Department of Translational Medicine, University of Eastern Piedmont and Azienda Ospedaliero—Universitaria Maggiore della Carità, 28100 Novara, Italyclara.deambrogi@med.uniupo.it (C.D.);

**Keywords:** diffuse large B cell lymphoma, monoclonal antibodies, bispecific antibodies, epcoritamab, glofitamab, mosunetuzumab, odronextamab

## Abstract

Bispecific monoclonal antibodies (BsAbs) represent a promising new treatment option for many hematological malignancies. BsAbs are monoclonal antibodies (mAbs) with dual affinity, generally targeting CD3 expressed by T lymphocytes and a cancer-associated antigen, enabling the redirection of the immune response against cancer cells. The treatment landscape for diffuse large B-cell lymphoma (DLBCL) has undergone significant changes over the past few decades and continues to evolve. BsAbs have emerged as an effective and well-tolerated treatment, particularly for relapsed or refractory (R/R) DLBCL. This review aims to provide an overview of the current and emerging clinical applications of BsAbs in the treatment of DLBCL.

## 1. Introduction

Diffuse large B-cell lymphoma (DLBCL) is an aggressive malignancy and the most prevalent subtype of non-Hodgkin lymphoma (NHL) worldwide, accounting for around 4 out of 10 of all lymphoma diagnoses. The estimated number of new DLBCL cases in 2025 is ~32.500 in the United States and ~28.000 in Western Europe, reflecting an increase over the past years due to population aging and typical onset of the disease after the age of sixty [1,2,3]. DLBCL is typically diagnosed de novo in previously healthy individuals, although a subset may arise from pre-existing indolent low-grade B-cell lymphoproliferative disorders [4].

The incorporation of the anti-CD20 monoclonal antibody (mAb) rituximab into conventional chemotherapy with cyclophosphamide, doxorubicin, vincristine, and prednisone (CHOP) significantly increased the number of complete responses, leading to a 10-year disease-free survival (DFS) of approximately 65% [5,6]. For more than 20 years, R-CHOP has represented the standard of care (SOC) as a frontline regimen in DLBCL. No other regimen, including intensified chemoimmunotherapy [7,8,9], second-generation anti-CD20 monoclonal antibody [10], maintenance therapy [11,12,13], or targeted therapies [14,15,16,17,18], has shown superiority over R-CHOP in terms of overall survival (OS) and progression-free survival (PFS). Recently, Pola-R-CHP, a modified R-CHOP regimen in which vincristine is replaced by polatuzumab vedotin, has been shown to reduce the risk of disease progression in previously untreated intermediate- or high-risk DLBCL [19]. However, 40% of patients experience refractory or relapsed (R/R) disease: primary refractory disease, accounting for 10–15% of cases, is defined as the failure to achieve a response to initial treatment, while relapsed disease (20–30% of cases) typically occurs within two years after completion of therapy and is characterized by the appearance of new lesions after achieving a complete response [20,21]. The prognosis of R/R DLBCL is particularly poor, especially in primary refractory disease or early relapse (within 3 to 6 months), with a median OS of approximately six months [22], thus representing a significant unmet clinical need.

The second line treatment for R/R disease has long been represented by salvage chemotherapy followed by autologous stem-cell transplantation (ASCT) for those who achieve an objective response, resulting in a durable remission in approximately 40–50% of patients [23,24,25]. However, high-dose conditioning chemotherapy and subsequent ASCT are associated with both acute and long-term treatment-related toxicities, and many patients are not eligible for such treatment due to age or preexisting comorbidities. Chimeric Antigen Receptor (CAR) T-cell therapy is an innovative form of cellular therapy in which a patient’s own lymphocytes are engineered to selectively target lymphomatous cells. Although it represents a breakthrough in the treatment of DLBCL, particularly in R/R disease, eligibility for CAR T-cell therapy remains limited by regulatory restrictions, age, comorbidities, and—at least in some contexts—cost. The advent of other novel therapeutic agents has provided alternative options to conventional treatments or CAR T-cell therapy, addressing the need for care for those considered frail or unfit for cellular therapy.

Bispecific monoclonal antibodies (BsAbs) are an innovative class of drugs designed to simultaneously bind two distinct antigens, enabling the activation of the immune system against tumor cells, the blockade of multiple signaling pathways, or the delivery of a cytotoxic payload to target cells [26]. Bispecific T-cell engagers (BiTE), dual affinity retargeting (DART) molecules and IgG-like bispecific antibodies are capable of redirecting T-cells against cancer cells. One arm of these molecules binds CD3 on T-cells, while the other targets a tumor-associated antigen on the surface of cancer cells (Figure 1). BiTEs are recombinant bispecific single-chain antibody constructs, structurally composed of two single-chain variable fragments (scFv) connected by a flexible linker, without a Fragment crystallizable (Fc) region (Figure 2) [27]. Blinatumomab is the first-in-class CD3 × CD19 BiTE approved for the treatment of hematological malignancies. However, the lack of an Fc region is associated with a short plasma half-life, as these molecules lack protection from catabolism by the neonatal Fc receptor (FcRn), requiring continuous infusion [28,29,30]. DARTs consist of two scFv arranged in a cross-bridged conformation, where the variable regions are linked by a disulfide bond (Figure 2). Some DARTs also include an IgG1 Fc region, which extends half-life and improves pharmacokinetics. This structural design enhances stability, affinity, and half-life compared to BiTEs. Finally, IgG-like bispecific antibodies retain the classical IgG structure, which improves pharmacokinetic properties, preserves Fc-mediated effector functions, and minimize the potential for immunogenicity [27]. This format is employed in most clinically approved BsAbs for B-cell malignancies, including epcoritamab, glofitamab, mosunetuzumab and odronextamab [31,32,33,34]. BsAbs can be administered via subcutaneous injection or intravenous infusion, depending on the specific molecule and clinical indication.

BsAbs are emerging as a novel and promising therapeutic approach for various hematological malignancies, particularly NHL and multiple myeloma, showing potential both as part of combination regimens in frontline treatment and in R/R disease [35]. Moreover, BsAbs may represent an efficient and safe option for patients who are not eligible for cellular therapies. In this review, we examine the current landscape of BsAbs for the treatment of DLBCL.

## 2. Epcoritamab

Epcoritamab (GEN3013) is an IgG1 CD3 × CD20 bispecific antibody, whose safety and efficacy in R/R large B-cell lymphoma (LBCL) was first evaluated in the pivotal EPCORE NHL-1 trial (NCT03625037). Epcoritamab is administered once weekly with a step-up dosing schedule during weeks 1 to 3 of cycle 1: a 0.16 mg priming dose on day 1, followed by a 0.8 mg intermediate dose on day 8, and then 48 mg as the full dose once weekly through cycle 3. Thereafter, treatment is given every two weeks from cycles 4 to 9 and every four weeks from cycle 10 onward, until disease progression or unacceptable toxicity [31]. The results of EPCORE NHL-1 trial led to the approval of epcoritamab by the U.S. Food and Drug Administration (FDA) and the European Medicines Agency (EMA) for patients with R/R DLBCL who had received at least two prior systemic treatments.

The most common treatment-emergent adverse events (AEs) were cytokine release syndrome (CRS) (51.0%), pyrexia (24.8%), fatigue (24.2%) and neutropenia (23.6%). Immune effector cell-associated neurotoxicity syndrome (ICANS) was reported as an AE of special interest in 6.4% of patients. Most of the reported CRS and ICANS cases were low-grade [1 or 2] (47.8% and 5.8%, respectively). Most CRS cases occurred after administration of the first full dose of epcoritamab on day 15 of cycle 1, the median onset time was 20 h and the median time to resolution was 48 h [31,36].

### 2.1. Epcoritamab in the Frontline Setting

The potential clinical use of epcoritamab in newly diagnosed DLBCL is under investigation in various clinical trials. The EPCORE NHL-2 trial (NCT04663347) is an ongoing phase Ib/II study to evaluate the safety and efficacy of epcoritamab in combination with other SOC agents in participants with B-NHL. In Arm 1, patients with a newly diagnosed IPI ≥ 3 DLBCL received epcoritamab + R-CHOP. At the time of reporting, the ORR was 100% (CR 87%), and the estimated 24-month PFS and OS rates were 74% and 87%, respectively. Notably, the CR rate in double-hit/triple-hit DLBCL was 83%, similar to the overall population [37].

These encouraging data supported further evaluation of epcoritamab + R-CHOP for the treatment of newly diagnosed DLBCL in the ongoing phase III EPCORE DLBCL-2 trial (NCT05578976) [38]. The EPCORE DLBCL-3 trial (NCT05660967) is evaluating the safety and efficacy of epcoritamab alone or in combination with lenalidomide in older or unfit patients with newly diagnosed DLBCL who are not eligible for anthracycline-based therapy. In the initial analysis of epcoritamab monotherapy, at a median follow-up of approximately 6 months, the ORR was 74% (CR 64%). The median time to response was 1.4 months, and the median time to CR was 2.4 months. Eighty-four percent of responders maintained their response, 88% of patients who achieved CR remained in CR, and 76% of all patients were still alive [39]. In Arm 8 of the EPCORE NHL-2 trial, patients with newly diagnosed DLBCL who were not eligible for full dose of R-CHOP received epcoritamab + R-mini-CHOP. The median age was 81 years, and approximately 70% of patients had an IPI ≥ 3. At a median follow-up of 16.8 months, the ORR was 89% (CR 82%), and the estimated 12-month PFS and OS rates were 88% and 96%, respectively. The safety profile was manageable, with no ICANS reported and only low-grade CRS events [40]. As shown in EPCORE NHL-1, patients who achieved CR had durable responses: among responders, 83% of patients remained in CR at 24 months in Arm 1, and 92% remained in CR at 12 months in Arm 8 [37,40,41].

Moreover, the ongoing phase Ib/II EPCORE NHL-5 trial (NCT05283720) is evaluating the safety and efficacy of epcoritamab in combination with antineoplastic agents in adults with NHL. Initial data from Arm 3, which included newly diagnosed DLBCL patients treated with epcoritamab + Pola-R-CHP, reported an ORR of 100% (CR 89%) at a median follow-up of 5.8 months, with a median time to response of 2.7 months [42]. Finally, the phase II NCT06045247 trial is evaluating the combination of epcoritamab and R-mini-CVP (rituximab, cyclophosphamide, vincristine and prednisone) in older or unfit patients who are ineligible for anthracycline-based therapy.

### 2.2. Epcoritamab in Relapsed/Refractory DLBCL

The long-term follow-up of the pivotal EPCORE NHL-1 trial reported a median PFS, OS, and duration of response (DoR) of 4.4, 18.5 and 17.3 months, respectively. The estimated 24-month PFS, OS and DoR rates were 27.8%, 44.6%, and 42.6%. At a median follow-up of approximately 25 months, the overall response rate (ORR) was 63.1% and the complete response (CR) rate was 40.1%. At 30 months, 54% of complete responders remained in CR. Among complete responders, the estimated 30-month PFS and OS rates were 55% and 71%, respectively [31,36,41].

The encouraging results of the EPCORE NHL-1 study led to the initiation of the ongoing phase III EPCORE DLBCL-1 trial (NCT04628494), which is comparing epcoritamab monotherapy versus SOC regimens, namely R-GemOx (rituximab plus gemcitabine and oxaliplatin) and BR (bendamustine plus rituximab), in patients with R/R DLBCL who had failed or were not eligible for ASCT [43].

The EPCORE NHL-2 trial includes several arms enrolling R/R DLBCL patients who are either eligible or ineligible for ASCT. In Arm 4, patients received either epcoritamab + R-DHAX/C (rituximab, dexamethasone, high-dose cytarabine and oxaliplatin/carboplatin) followed by ASCT if they achieved an objective response to salvage treatment, or epcoritamab alone until disease progression if ASCT was deferred. This cohort included patients with progressive disease (83%), primary refractory disease (66%), and prior CAR T-cell exposure (10%). At a median follow-up of approximately 28 months, 55% of patients proceeded to ASCT, while 7% remained on treatment. The ORR was 76% (CR 69%) with a median time to response of 1.4 months. The estimated 24-month PFS and OS rates were 60% and 86%, respectively; 90% of patients who proceeded to ASCT and 60% of those who continued epcoritamab without ASCT remained progression-free [44]. In Arm 5, patients with R/R DLBCL who were ineligible for ASCT received epcoritamab + GemOx. This cohort included patients with challenging-to-treat disease: most patients (62%) were treated with two or more lines of therapy, 52% had primary refractory disease, 70% were refractory to last therapy and 28% were previously treated with CAR T-cell. At a median follow-up of 13.2 months, the ORR was 85% with a CR rate of 61%, the median duration of CR and the median OS were 23.6 and 21.6 months, respectively [45].

The efficacy of epcoritamab + lenalidomide compared to epcoritamab + R-GemOx in patients with R/R DLBCL who had failed or were not eligible for ASCT or CAR T-cell will be evaluated in the EPCORE DLBCL-4 trial (NCT06508658) [46]. The EPCORE NHL-5 trial includes several arms (Arm 1, 2, 4) designed to evaluate the safety and efficacy of epcoritamab in combination with antineoplastic agents in adults with R/R DLBCL. Patients in Arm 1 received epcoritamab + lenalidomide, those in Arm 2 received epcoritamab + lenalidomide + ibrutinib, and those in Arm 4 received epcoritamab + CC-99282 (CELMoD). The first results from Arm 1 showed that, at a median follow-up of 11.5 months, the ORR was 67.6% (CR 51.4%). The median DoR and the median duration of CR were not reached (NR) [47].

The NCT06287398 trial has been designed to evaluate epcoritamab + R-DHAOx (rituximab, dexamethasone, high-dose cytarabine and oxaliplatin), followed by ASCT and consolidation with epcoritamab alone in R/R DLBCL. Moreover, the EpLCART trial (NCT06414148) is designed to evaluate the efficacy of epcoritamab as a single agent or epcoritamab + R2 (rituximab and lenalidomide) as a consolidation treatment after anti-CD19 CAR T-cell therapy for patients who responded to treatment and are at high risk of progression.

Finally, the safety and efficacy of other epcoritamab-based combinations are currently under investigation in several clinical trials, such as epcoritamab + R-GDP (rituximab, gemcitabine, dexamethasone and cisplatin) in transplant- or CAR T-cell eligible patients in the phase II NCT05852717 study [48], or epcoritamab administered before and after CAR T-cell therapy in the phase II NCT06458439 trial. Results of clinical trials involving epcoritamab are summarized in Table 1.

## 3. Glofitamab

Glofitamab (RG6026) is an IgG-like CD3 × CD20 bispecific antibody with a unique 2:1 molecular structure, enabling bivalent binding to CD20 and monovalent binding to CD3 [49]. The results of the phase I/II NP30179 trial (NCT03075696), which evaluated glofitamab monotherapy in R/R NHL, led to its approval by both the FDA and EMA for patients with R/R DLBCL who had received at least two prior lines of therapy [32]. More recently, based on the findings of the phase III STARGLO trial (NCT04408638), the combination of glofitamab + GemOx was approved by EMA for patients with R/R DLBCL who are ineligible for ASCT [50]. Glofitamab is administered intravenously using a step-up dose schedule. On day 1 of cycle 1, patients receive a pre-treatment of 1000 mg obinutuzumab, followed by glofitamab at 2.5 mg on day 8 and 10 mg on day 15 of cycle 1. The full dose of 30 mg is then administered from day 1 of cycle 2. Treatment continues until disease progression or unacceptable toxicity, for a maximum of 12 cycles [32]. The safety and efficacy of the subcutaneous formulation are currently under investigation in the phase Ib BP43015 study (ISRCTN17975931; 2021-000064-29).

The most common treatment-emergent AEs were CRS (63%), neutropenia (38%), anemia (31%), and thrombocytopenia (25%); ICANS occurred in 8% of patients. Most reported cases of CRS and ICANS were low-grade [1 or 2] (59% and 5%, respectively), and CRS typically developed during the first three administration of glofitamab. The median time to onset from day 8 of cycle 1 was 13.5 h, with a median duration of 30.5 h [32]. Currently, glofitamab is under investigation in several DLBCL clinical trials, both in the frontline and relapsed/refractory settings, mainly in combination with chemoimmunotherapy and other innovative drugs.

### 3.1. Glofitamab in the Frontline Setting

The NP40126 trial (NCT03467373) is a phase Ib study designed to evaluate glofitamab administered in combination with obinutuzumab (G), rituximab (R) and standard dose CHOP (G/R-CHOP or R-CHOP) in participants with R/R NHL, and in combination with G/R-CHOP or Pola-R-CHP in participants with newly diagnosed DLBCL. In the latter setting, after a median follow-up of approximately 5 months, the best ORR was 100% (CR 76.5%) for glofitamab + Pola-R-CHP. For glofitamab + R-CHOP, after a median follow-up of 17.1 months, the ORR was 92.9% (CR 83.9%), and the median time to CR was 1.8 months; among complete responders, the estimated 12-month probability of remaining in CR was 91.5%, and the median DoR was NR [51,52].

The safety and tolerability of glofitamab in combination with R-CHOP (Arm A) or Pola-R-CHP (Arm B) for younger patients (18–65 years) with high-risk DLBCL or high-grade B-cell lymphoma (HGBL) is currently being investigated in the ongoing phase Ib/II COALITION study (NCT04914741). At a median follow-up of 14.6 months, the best ORR was 100% for the entire cohort, with a CR rate of 92%. In Arm A, at end of induction (EOI), CR and partial response (PR) rates were 70% and 29%, respectively; in Arm B they were 80% and 18%. The 12-month PFS and OS rates were 88% and 96% in Arm A and 95% and 97% in Arm B, respectively [53]. Similarly, glofitamab in combination with Pola-R-CHP is being investigated in the phase II NCT05800366 trial in patients with newly diagnosed high-risk DLBCL. In light of these results, the phase III SKYGLO study (NCT06047080) has been initiated to evaluate the efficacy and safety of glofitamab + Pola-R-CHP versus Pola-R-CHP in patients with newly diagnosed LBCL [54].

The NCT04980222 trial is evaluating the safety and efficacy of glofitamab + R-CHOP in LBCL defined as high-risk by circulating tumor DNA (ctDNA) dynamics. Preliminary results showed an ORR of 93.3% and a CR rate of 80% at EOT [55]. An adaptive approach incorporating ctDNA and PET to optimize frontline treatment for DLBCL is being evaluated in the phase II GRAIL study (NCT06050694). After two initial cycles of Pola-R-CHP, ctDNA low-risk patients (favorable ctDNA and PET2) will receive two additional cycles of Pola-R-CHP chemotherapy followed by two additional courses of rituximab. Conversely, ctDNA high-risk patients (unfavorable ctDNA and/or PET2) will receive four additional cycles of Pola-R-CHP chemotherapy in combination with glofitamab. The combination of glofitamab with polatuzumab vedotin and rituximab is being investigated in the NCT05798156 trial for older or unfit patients with newly diagnosed aggressive B-cell lymphomas who are not eligible for the full dose of R-CHOP. The initial results demonstrated a manageable toxicity profile in this difficult-to-treat population [56]. Similarly, the GLORY trial (NCT06765317) has been designed to evaluate the efficacy of glofitamab + Pola-R-mini-CHP with a PET-guided approach in previously untreated older or frail patients with DLBCL or HGBCL.

### 3.2. Glofitamab in Relapsed/Refractory DLBCL

Several trials are currently investigating the role of glofitamab in improving outcomes for patients with R/R DLBCL, mainly in combination with other therapeutic regimens. According to the long-term results of the pivotal NP30179 trial, the median PFS, OS, and DoR were 4.9, 11.5 and 18.4 months, respectively, while the 12-month PFS, OS, and DoR rates were 37%, 50% and 64%, respectively. At an extended follow-up of over three years, the ORR was 52% and CR rate was 39%. Among complete responders, the estimated PFS and OS rates were 57% and 77%, respectively. The median duration of CR was 29.8 months [32,57].

The phase III STARGLO trial (NCT04408638) evaluated the efficacy and safety of glofitamab plus GemOx or R-GemOx in R/R DLBCL patients not eligible for ASCT. This study demonstrated improved OS and PFS with glofitamab plus GemOx, albeit with increased but manageable toxicity. At a median follow-up of 20.7 months, the ORR was 69.9% (CR 57.4%) versus 37.4% (CR 23.1%) for glofitamab + GemOx and R-GemOx, respectively (*p* < 0.001). The median OS and PFS were 25.5 and 14.4 months, respectively, both significantly higher with glofitamab + GemOx (both *p*-value < 0.001); the median duration of CR was NR. The 24-month OS rate was 52.8% and the 12-month PFS rate was 53.2% [50]. The NCT05364424 trial is a phase Ib study evaluating glofitamab in combination with R-ICE (rituximab, ifosfamide, carboplatin, and etoposide) in patients with R/R DLBCL who are eligible for either transplantation or CAR T-cell therapy. At the time of the interim analysis, the best ORR was 78.1% (CR 68.8%) [58]. The NP39488 trial (NCT03533283) is a phase Ib/II study investigating the combination of glofitamab with atezolizumab or polatuzumab vedotin in patients with R/R NHL. In the polatuzumab vedotin arm, at a median follow-up of 23.5 months, the best ORR was 80% (CR 62%) across all histologies and 84% (CR 61%) in patients with DLBCL. The median PFS, OS, and DoR were 12.3, 39.2 and 21.9 months, respectively, for the entire cohort, while the median PFS was 10.4 months in patients with DLBCL [59]. In the atezolizumab arm, the ORR was 36% and CR rate 17% across all histologies, and the ORR was 29% and the CR rate was 9.7% for aggressive NHL [60].

The combination of englumafusp alfa (RO7227166), an antibody-like fusion protein that simultaneously targets CD19 on B cells and 4-1BB on T-cells and other immune cells, with glofitamab in R/R NHL, including DLBCL, is being evaluated in the phase I/II NCT04077723 study. Extended follow-up results from the dose-escalation part of phase I demonstrated a best ORR of 67.5% (CR 56.6%) in aggressive NHL. The median DoR, PFS, and OS were 31.5, 9.7 and 20.9 months, respectively, with a 1-year PFS rate of 45.6% [61]. Similarly, the phase I NCT05219513 study investigated the combination of glofitamab and RO7443904, a CD19 × CD28 bispecific antibody, in patients with R/R NHL. In the presence of a T-cell receptor signal, RO7443904 provides T-cells co-stimulation via CD28 agonism, thereby enhancing T-cell activation, proliferation, effector functions, and potentially reversing T-cell exhaustion. Preliminary data demonstrated an ORR of 64% with a CR rate of 39% for aggressive NHL [62]. The phase II GPL study (NCT05335018) is evaluating the combination of the BTK inhibitor poseltinib with glofitamab and lenalidomide in patients with R/R DLBCL. Interim analysis demonstrated an ORR of 89.3% with a CR rate of 42.9%. The 6-month OS, PFS, and DoR rates were 81%, 55% and 66%, respectively [63].

Finally, several clinical trials are evaluating the efficacy and safety of novel glofitamab-based combinations in R/R DLBCL, including glofitamab + maplirpacept (NCT05896163), an anti-CD47 fusion protein designed to enhance phagocytosis and antitumor activity by inhibiting CD47-mediated signaling [64]; glofitamab + CELMoD (NCT05169515); glofitamab + axicabtagene ciloleucel (NCT06213311) [65]; glofitamab with or without lenalidomide and hypofractionated radiotherapy (NCT06867536, NCT06651853); glofitamab + tucidinostat (NCT06570447), an HDAC inhibitor; glofitamab + R-GDP (NCT04161248); glofitamab as a bridge to ASCT (NCT06682130) or as sequential therapy with CAR T-cell treatment (NCT04889716); and loncastuximab tesirine + glofitamab in arm E of the LOTIS-7 trial (NCT04970901) [66]. Results of clinical trials involving glofitamab are summarized in Table 2.

## 4. Odronextamab

Odronextamab (REGN1979) is a fully human, hinge-stabilized IgG4 CD3 × CD20 bispecific antibody that has shown significant therapeutic activity in NHL. The pivotal ELM-1 trial (NCT02290951) was the first to evaluate the safety and efficacy of odronextamab in patients with R/R NHL, reporting encouraging results in both R/R follicular lymphoma (FL) and DLBCL [34]. Odronextamab has received EMA approval for the treatment of R/R DLBCL and R/R FL patients who have received at least two prior lines of therapy. It is administered intravenously using a step-up dosing regimen to mitigate immune-related AEs. Specifically, during the first cycle, odronextamab is given as a 4 h infusion twice weekly (0.2 mg on day 1, 0.5 mg on day 2, 2 mg on days 8 and 9, and 10 mg on days 15 and 16). From cycles 2 to 4, it is administered once weekly at an increased dose of 160 mg. The treatment then continues with a maintenance phase of 320 mg every two weeks, extended to every four weeks if a patient maintains a complete response for at least 9 months [34].

The most common treatment-emergent AEs of any grade were CRS (52.9%), anemia (35.8%), pyrexia (39.6%), and neutropenia (31.6%). Among patients who received odronextamab using the step-up dosing regimen, 49.0% experienced CRS events, most of which were low-grade (grade 1: 33%; grade 2: 15%), with only 1% experiencing grade 3 events. No ICANS were reported. CRS was predominantly confined to cycle 1 [34]; the median time to onset was 17.8 h and the median duration time was 7.1 h [69].

### 4.1. Odronextamab in the Frontline Setting

Based on the results of the ELM-2 trial, which demonstrated the efficacy of odronextamab in R/R DLBCL, the phase III OLYMPIA-3 trial (NCT06091865) has been initiated to evaluate odronextamab in combination with CHOP versus R-CHOP in patients with newly diagnosed DLBCL [70]. Additionally, several ongoing clinical trials are assessing the efficacy of odronextamab-based combinations in patients with R/R DLBCL. These include the ATHENA-1 study (NCT05685173), which evaluates odronextamab in combination with REGN5837, a CD28 × CD22 IgG4-based bispecific antibody designed to provide co-stimulatory signaling [71]; the CLIO-1 study (NCT02651662), investigating odronextamab combined with cemiplimab, a monoclonal antibody targeting programmed cell death protein 1 (PD-1) [72]; and a trial (NCT06854159) investigating the administration of odronextamab before and after CAR T-cell therapy.

### 4.2. Odronextamab in Relapsed/Refractory DLBCL

The primary analysis of the ELM-1 expansion cohort demonstrated that, in the prespecified subgroup of patients with R/R DLBCL previously exposed to CAR T-cell therapy, at a median follow-up of 16.2 months, the median PFS, OS, and DoR were 4.8, 10.2 and 14.8 months, respectively; the ORR was 48% and the CR rate was 32% [73].

The phase II ELM-2 trial (NCT03888105), which included patients with R/R DLBCL without prior CAR T-cell exposure, demonstrated that, at a median follow-up of 26.2 months, the median PFS, OS, and DoR were 4.4, 9.2, and 10.2 months, respectively; the ORR was 52% and the CR rate was 31.5%. The estimated 24-month PFS, OS, and DoR rates were 21.1%, 31.6%, and 36.9%, respectively [74,75]. In a pooled analysis of the ELM-1 and ELM-2 studies, at a median follow-up of 23.0 months, the median PFS, OS, and DoR were 4.4, 9.3, and 10.5 months; the ORR was 50.8% and CR rate was 31.6%. The probability of maintaining CR at 36 months was 51.0%, while the estimated 36-month PFS and OS rates were 17.5% and 27.0%, respectively. According to these results, odronextamab monotherapy demonstrated a manageable safety profile and encouraging preliminary activity, including durable responses in heavily pretreated patients with B-cell non-Hodgkin lymphoma, supporting further clinical investigation [75]. The results of clinical trials involving odronextamab are summarized in Table 3.

## 5. Mosunetuzumab

Mosunetuzumab is a full-length, humanized IgG1-like CD3 × CD20 bispecific antibody. It has been investigated as a new therapeutic option in both indolent and aggressive NHL, mainly in the R/R setting. The pivotal phase I GO29781 (NCT02500407) trial assessed the safety and efficacy of mosunetuzumab as monotherapy or in combination with atezolizumab in patients with R/R NHL, including DLBCL. Mosunetuzumab, in its intravenous formulation, has been approved by FDA and EMA for the treatment of R/R FL after at least two previous lines of therapy, whereas it is currently not approved for DLBCL [33]. It is administered intravenously in 21-day cycles with a step-up dosing schedule (1 mg on day 1 of cycle 1; 2 mg on day 8 of cycle 1; 60 mg on day 15 of cycle 1 and on day 1 of cycle 2; 30 mg from day 1 of cycle 3 onwards). The schedule consists of 8 cycles if a CR is achieved, whereas the duration can be extended up to 17 cycles in case of PR or stable disease (SD) by cycle 8 [76]. Primary results in R/R FL demonstrated the non-inferiority of subcutaneous administration compared with intravenous, using a 21-day step-up dosing schedule: 5 mg on day 1 of cycle 1; 45 mg on days 8 and 15 of cycle 1; and 45 mg from day 1 of cycle 2 onward. Patients achieving CR by cycle 8 completed treatment, whereas those with PR or SD continued therapy for up to 17 cycles [77].

The most common treatment-emergent AEs were neutropenia (27.3%), CRS (26.1%), fatigue (26.1%) and rash (21.6%); ICANS occurred in 1.1% of patients. Most CRS cases were low-grade [1 or 2] (20.5% and 3.4%, respectively). CRS predominantly occurred during cycle 1, particularly on day 15 of the step-up dosing schedule, with a median duration of 26.0 h and a median resolution time of 3 days [76].

### 5.1. Mosunetuzumab in the Frontline Setting

Several clinical trials are evaluating the efficacy of mosunetuzumab as a first-line treatment in patients with newly diagnosed DLBCL. The phase Ib/II GO40515 (NCT03677141) trial investigated mosunetuzumab in combination with CHOP (M-CHOP) and Pola-CHP (Pola-M-CHP) in patients with newly diagnosed DLBCL. In the M-CHOP arm, the ORR and CR rates were 87.5% and 85.0%, respectively. At a median follow-up of 32 months, the estimated 2-year PFS rate was 65.4%. In this study, the M-CHOP regimen emerged as a promising frontline treatment option for patients with newly diagnosed DLBCL [78]. On these grounds, the GO40515 trial compared Pola-M-CHP versus Pola-R-CHP. The median PFS, OS, and DoR were not estimable for either regimen. The estimated 24-month PFS, OS, and DoR rates were 70.8%, 86.3%, and 71.4% for Pola-M-CHP regimen, compared with 81.8%, 86.4%, and 80.7% for Pola-R-CHP. The ORR and CR rates were 75% and 72.5% for Pola-M-CHP, respectively, whereas they were 86.4% and 77.3% for Pola-R-CHP. Based on these results, the Pola-M-CHP regimen did not demonstrate a clear efficacy advantage over Pola-R-CHP in patients with newly diagnosed DLBCL neither in terms of PFS (*p* = 0.66), event-free survival (EFS) (*p* = 0.48) nor terms of OS (*p* = 0.97) [79].

The phase II GO40554 study (NCT03677154) investigated the role of mosunetuzumab in patients with previously untreated DLBCL. In cohort A, mosunetuzumab was administered as consolidation therapy in patients who achieved SD or PR after standard frontline immunochemotherapy. Cohorts B and C focused on elderly or unfit patients with previously untreated DLBCL. In cohort B mosunetuzumab was administered intravenously as monotherapy, whereas in cohort C it was given subcutaneously in combination with polatuzumab vedotin. In cohort B, at a median follow-up 23.3 months, the best ORR and CR rates were 56% and 43%, respectively, whereas the ORR and CR rates at the EOT were 43% and 35%. The median duration of CR was 15.8 months and the estimated 12-month PFS rate was 39% [80]. In cohort C, at a median follow-up of 7.5 months, the best ORR and CR rates were 80% and 61%, respectively, whereas the ORR and CR rates at the EOT were 55% and 45%. The median PFS was 11.9 months and the estimated 12-mont PFS rate was 49.7% [81]. Similar to cohort A of the GO40554 study, the phase II NCT06828991 study has been designed to evaluate the impact of mosunetuzumab as consolidation therapy in elderly or unfit patients newly diagnosed with DLBCL who were treated with Pola-R-mini-CHP and have detectable ctDNA following induction therapy. Finally, the phase II NCT06594939 trial is evaluating subcutaneous mosunetuzumab in combination with a split dose of Pola-CHP regimen in elderly patients with newly diagnosed with DLBCL.

### 5.2. Mosunetuzumab in Relapsed/Refractory DLBCL

The pivotal GO29781 study demonstrated a median PFS, OS, and DoR of 3.2, 11.5, and 7.0 months, respectively, in patients with R/R DLBCL. The estimated 12-month PFS, OS, and DoR rates were 22.6%, 48.1%, and 44.1%, respectively. The ORR was 42% and CR rate was 23.9% at a median follow-up of 10.1 months. Among complete responders, the median duration of CR was NR.

The phase Ib/II GO40516 study (NCT03671018) is evaluating the safety and efficacy of both subcutaneous and intravenous mosunetuzumab in combination with polatuzumab vedotin (M-Pola) in patients with R/R NHL. The primary analysis showed a median PFS, OS, and DoR of 11.4, 23.3, and 20.8 months, respectively, for intravenous M-Pola in R/R LBCL. The estimated 24-month PFS, OS, and DoR rates were 31.3%, 48.6%, and 49.7%. At a median follow-up of 23.9 months, the best ORR and CR rates were 59.2% and 45.9%. At 24 months, among complete responders, 60.8% are estimated to remain in CR [82]. Based on the results of the dose-expansion phase, the contribution of subcutaneously administered mosunetuzumab to the M-Pola combination was evaluated in comparison to rituximab plus polatuzumab vedotin (R-Pola) in the phase II portion of the study. At a median follow-up of 18 months, the median PFS, OS, and DoR for M-Pola were NR, whereas for R-Pola they were 6.4, NR and 10.1 months, respectively. For M-Pola versus R-Pola, the best ORR was 78% versus 58%, and the CR rate was 50% versus 35%, respectively [83]. These encouraging data supported further evaluation of M-Pola in the ongoing phase III SUMNO trial (NCT05171647) [84].

Finally, several clinical trials are currently evaluating the efficacy and safety of other mosunetuzumab combinations in patients with R/R DLBCL, including mosunetuzumab as consolidation therapy following ASCT (NCT05412290) or sequential treatment CAR T-cell therapy (NCT04889716); mosunetuzumab + polatuzumab vedotin + lenalidomide (NCT06015880); mosunetuzumab + polatuzumab vedotin + tafasitamab + lenalidomide (NCT05615636); CAR T-cell + mosunetuzumab + polatuzumab vedotin (NCT05260957); mosunetuzumab in combination with tiragolumab, an anti-TIGIT agent, with or without atezolizumab (NCT05315713); mosunetuzumab + CELMoD (NCT05169515); loncastuximab tesirine + mosunetuzumab in arm F of LOTIS-7 trial (NCT04970901) [66] and in NCT05672251 study [85]. Results of clinical trials involving mosunetuzumab are summarized in Table 4.

## 6. Bispecific Antibodies: Similarities and Differences

At the time of writing this manuscript, epcoritamab and glofitamab as monotherapy had received FDA and EMA approval for the treatment of R/R DLBCL starting from the third line of therapy; more recently, EMA approved glofitamab in combination with GemOx for the treatment of patients with R/R DLBCL who are ineligible for ASCT. This combination has not received FDA approval due to concerns regarding the enrollment of the STARGLO trial and the applicability of its results to the US population. In fact, of the 274 patients enrolled in the intention-to-treat population, 88 were from Europe, 161 from Asia or Australia and 25 from North America. In these subgroups, glofitamab plus GemOx compared to R-GemOx showed different OS benefits in European (HR 1.09; 95% CI 0.54–2.18), Chinese and Australian (HR 0.41; 95% CI 0.27–0.64), and North American (HR 2.62; 95% CI 0.56–12.34) [50]. Odronextamab as monotherapy has received EMA approval for the treatment of R/R DLBCL and R/R FL after the second line of therapy. Mosunetuzumab is approved for R/R FL after at least two prior lines, and its role in DLBCL is currently being investigated in several clinical trials.

Although epcoritamab, glofitamab, mosunetuzumab and odronextamab are all IgG-like CD3 × CD20 BsAbs, they differ in their molecular construct, resulting in different pharmacokinetics, T-cell recruiting efficiency, routes of administration and safety profile. Epcoritamab, mosunetuzumab and odronextamab bind monovalently to CD20, whereas glofitamab binds bivalently, which may enhance avidity but also increase the risk of cytokine release. Epcoritamab is administered subcutaneously, while glofitamab, mosunetuzumab, and odronextamab are administered intravenously. Subcutaneous formulation of glofitamab and mosunetuzumab are currently under investigation. Epcoritamab and odronextamab are administered until disease progression or unacceptable toxicity, whereas glofitamab and mosunetuzumab are given as fixed-duration therapies. A step-up dosing approach is used to mitigate the risk of CRS in all schedules of these BsAbs (see respective paragraphs for details) [31,32,33,34,49,76].

The key details of the discussed BsAbs, including CRS and ICANS, are reported in Table 5.

## 7. CRS and ICANS

Bispecific antibodies are generally manageable and well tolerated; however, they are associated with specific toxicities, namely CRS and ICANS. As a consequence of T-cell activation induced by T-cell-engaging immunotherapies, proinflammatory cytokines—particularly IL-6, IL-1, IFN-γ and TNF-α—are released by T-cells and other immune effector cells. An excessive and supraphysiologic response to these cytokines defines CRS, a systemic inflammatory response characterized by fever, hypotension, tachycardia, and, in severe cases, multi-organ dysfunction. Furthermore, excessive activation of immune effector cells and endothelial dysfunction amplify the inflammatory cascade, potentially leading to capillary leak syndrome (CLS) and disseminated intravascular coagulation (DIC). The proinflammatory state, endothelial dysfunction, and increased blood–brain barrier permeability that may occur as a consequence of CRS, can contribute to the pathogenesis of neuroinflammation, which underlies ICANS. The latter is characterized by a spectrum of neurological manifestations, including altered consciousness, disorientation, aphasia, seizures, cerebral edema, and, in the most severe cases, coma. CRS and ICANS are usually reversible, without permanent sequelae [86,87].

The risk of CRS is influenced by the type of therapy, the underlying disease and patient-specific characteristics. In particular, disease burden and the so-called ‘first-dose effect’ are among the most important predictors of CRS. CRS manifestations most often develop after the first administration of BsAbs and typically do not recur thereafter. The ‘first-dose effect’ is thought to be associated with the high burden of disease at treatment initiation. The incidence of CRS in patients receiving BsAbs depends on the type of monoclonal antibody (Table 5). In most cases, CRS manifestations include flu-like symptoms, fever, fatigue, headache, arthralgia, myalgia and rash. Severe cases are characterized by high fever with hypotension and/or hypoxia that might require fluid challenge, vasopressor or oxygen therapy. In some cases, severe CRS is complicated by renal failure or signs of cardiac dysfunction with reduced ejection fraction. Rarely, life-threating complications such as DIC, CLS, acute respiratory distress syndrome (ARDS), hemophagocytic lymphohistiocytosis (HLH) or macrophage activation syndrome (MAS) have been reported [86]. The development and evolution of CRS can be diagnosed and monitored using clinical and laboratory parameters. C-reactive protein (CRP) is an acute-phase reactant primarily produced in response to IL-6, and its levels serve as a surrogate marker for IL-6 bioactivity. Therefore, monitoring CRP levels during BsAbs administration allows for the assessment of CRS evolution. Elevation of ferritin levels and hypertriglyceridemia have been observed in many patients who develop CRS, particularly in those with HLH-associated CRS, resembling the laboratory findings of HLH/MAS [86,88].

CRS and ICANS adverse events by bispecific antibody are briefly summarized in Table 6.

CRS grading is based on the severity of signs and symptoms and the extent of required treatment [89]. The treatment of CRS in the context of BsAbs is largely based on the experience with CAR T-cell therapy [86,90]. Since infectious diseases may mimic CRS, blood cultures, other infectious tests, along with empirical broad-spectrum antibiotics are recommended for CRS of any grade. Symptomatic management depends on the severity of the syndrome, ranging from mild measures (e.g., antipyretics, fluids) to intensive interventions (e.g., vasopressors, intubation) in intensive care unit (ICU). In moderate to severe cases, intravenous administration of the IL-6R-blocking antibody tocilizumab is indicated and corticosteroids (e.g., dexamethasone, methylprednisolone) may be considered, particularly in the presence of concomitant ICANS [90]. The development and severity of CRS do not appear to be associated with clinical response: patients who achieve complete remission (CR) may not experience CRS, while those who develop CRS may not reach CR [86].

Pre-existing neurological diseases, tumor burden, central nervous or leptomeningeal involvement are risk factors for the development of ICANS [90]. In most cases, ICANS is preceded by CRS manifestations. Therefore, CRS can be considered an ‘initiating event’ or a cofactor of ICANS. Generally, ICANS develops when CRS symptoms have subsided, although, less frequently, CRS and ICANS can co-occur. The incidence of ICANS in patients receiving BsAbs depends on the type of monoclonal antibody (Table 5). Generally, ICANS symptoms include word-finding difficulty, confusion, aphasia, impaired fine motor skills, and somnolence. In severe cases, seizures, cerebral edema, and coma may occur [87]. The Immune Effector Cell-Associated Encephalopathy (ICE) score is a tool used to assess patients for encephalopathy by evaluating multiple cognitive domains (orientation, naming, following commands, writing, attention). The severity of ICANS is determined by the assessment of the ICE score, along with the evaluation of other neurological domains, such as level of consciousness, motor symptoms, seizures, and signs of elevated intracranial pressure (ICP)/cerebral edema, which may occur with or without encephalopathy [89]. Low-grade ICANS is managed with diagnostic work-up (cross-sectional imaging, electroencephalography, and cerebrospinal fluid analysis) and supportive care, whereas severe ICANS is treated with corticosteroids [87,90]. Tocilizumab does not appear to be effective in the treatment of ICANS, likely due to the different pathophysiology of ICANS compared with CRS and its limited penetration across the blood–brain barrier [87].

## 8. Treatment Sequencing in the CAR T-Cell Era

The advent of novel therapies for the management of patients with DLBCL, particularly CAR T-cell therapy and BsAbs, has raised important questions regarding the role of hematopoietic stem cell transplantation (HSCT) and the optimal sequencing of these treatments in the clinical practice. Prior to the introduction of BsAbs and CAR T-cell therapy, patients with DLBCL who relapsed or progressed following first-line chemoimmunotherapy were typically treated with second-line chemotherapy [6,19], followed by high-dose chemotherapy conditioning and ASCT [23,24], achieving durable remissions in approximately 40–50% of patients [23,24,25,91]. However, prognosis remains poor in high-risk patients, such as those with primary refractory disease, early relapse, a high age-adjusted International Prognostic Index (aaIPI), or HGBCL [92,93]. CD19-directed CAR T-cell therapy has revolutionized the treatment of R/R DLBCL, achieving durable remission rates of up to 40%, even in patients with primary refractory disease or those who relapse after ASCT [94,95,96]. Furthermore, based on the results from the CORAL and LY.12 trials, which compared CAR T-cell therapy with standard treatment [19,91], as well as from studies investigating CAR T-cell therapy as a first-line salvage treatment instead of chemotherapy followed by ASCT [97,98,99], the possibility of using CAR T-cell therapy as second-line treatment has gained increasing attraction. The feasibility of administering CAR T-cell therapy to patients with R/R DLBCL after first-line treatment, as demonstrated by the PILOT study [100], has expanded therapeutic options for patients who are ineligible for ASCT due to advanced age or pre-existing comorbidities, conditions commonly associated with significant acute and long-term toxicities [23,101,102]. Overall, these studies highlight that CAR T-cell therapy represents a valid second-line treatment option for patients with high-risk R/R disease and for that ineligible for ASCT [23,100,101,102]. However, despite an ORR of 80% with second-line treatment CAR T-cell therapy, approximately 30% of patients do not achieve a CR, and among those who do, around 36% experience disease progression within one year [97,103,104].

Consensus guidelines support the use of ASCT in patients who relapse >12 months after completing first-line chemotherapy and are deemed fit enough to proceed with ASCT [105]. This approach offers a potentially curative option for approximately 40% of patients [25]. The efficacy of CAR T-cell therapy after ASCT failure has been demonstrated in the ZUMA-1, TRANSCEND NHL-001, and JULIET trials, which reported ORR ranging from 52% to 78% [95,106,107]. Conversely, limited data are available regarding salvage ASCT after CAR T-cell failure. A study conducted by Spiegel et al. reported discouraging outcomes for salvage therapies in patients who progressed after axi-cel therapy, with an ORR of only 29% and a median PFS of 55 days [103]. Therefore, CAR T-cell therapy remains a valid third-line option in patients who relapse following ASCT, while the reverse sequence appears less effective.

In this complex therapeutic landscape, bispecific antibodies represent a novel therapeutic option for patients with R/R DLBCL. BsAbs have demonstrated efficacy even in heavily pretreated populations with a poor prognosis, including patients who relapse after CAR T-cell therapy [31,32]. Although the efficacy of BsAbs after CAR T-cell failure has not been extensively studied, there is evidence of their role in this difficult-to-treat population. Epcoritamab has demonstrated efficacy in a subset of patients previously treated with CAR T-cell therapy, as reported by Thieblemont et al. In this subgroup, the ORR was 54.1% (CR 34.3%) and the median DoR was 9.7 months [31]. Similarly, glofitamab initially demonstrated efficacy in this population, with an ORR ranging from 61% to 67% (CR from 33% to 44.4%) in small-cohort retrospective studies [108,109]. These findings have been recently confirmed by the phase II BiCAR study (NCT04703686), which investigated the role of glofitamab in R/R DLBCL after CAR T failure. The best ORR was 76.1% (best CR 45.7%) and the median OS and PFS were 14.7 and 3.8 months, respectively, after a median follow-up of 15.3 months [68]. Moreover, Rentsch et al. retrospectively analyzed the kinetics of CAR T-cell-specific DNA before, during, and after glofitamab treatment in a small cohort of patients. CAR T-cell expansion was observed in three out of nine patients at a median of 35 days following glofitamab initiation [109]. The role of odronextamab in R/R DLBCL patients previously exposed to CAR T-cell was investigated in the expansion cohort of the ELM-1 study (NCT02290951), achieving an ORR of 48% (CR 32%) and a median PFS of 4.8 months [73]. Mosunetuzumab has also shown activity in a phase I/II study (NCT02500407) including patients with R/R NHL (90% of whom had LBCL) after CAR T-cell therapy, achieving an ORR of 40% (CR 23%) and a median PFS of 6.1 months among responders. Notably, a longer interval between CAR T-cell therapy and mosunetuzumab initiation was associated with higher response rates [110]. Recent real-world data indicate that the timing of relapse after CAR T-cell therapy is a strong predictor of response to subsequent BsAb treatment (PFS: HR 1.73 (1.18–2.54), *p* = 0.005; OS: HR 2.31 (1.40–3.82), *p* = 0.001). In a multicenter cohort of 92 LBCL patients treated with BsAbs after CAR T-cell failure, those who relapsed within three months had a significantly lower ORR (29%) and CR (10%) and a shorter median PFS (2.2 months) and OS (4.2 months), compared with patients relapsing between four and six months (ORR 54%, CR 25%, median PFS 3.7 months, median OS 9.1 months) or more than six months after CAR T-cell therapy (ORR 60%, CR 45%, median PFS 10.5 months, median OS NR). Furthermore, patients receiving BsAbs as first salvage therapy after CAR T-cell failure achieved better outcomes than those receiving them in later lines (median PFS/median OS NR vs. 2.7/9.1 months) (PFS: HR 2.33 (1.36–3.98), *p* = 0.002; OS: HR 2.41 (1.28–4.54), *p* = 0.006) [111]. Conversely, limited data are available regarding the efficacy of CAR T-cell therapy following BsAb treatment, as no prospective CAR T-cell trials have included patients previously treated with BsAbs [31,32]. Furthermore, potential limitations related to the tumor microenvironment could impact CAR T-cell therapy outcomes after BsAb administration [112]. Continuous exposure to BsAbs could lead to T-cell exhaustion, potentially reducing CAR T-cell proliferation and efficacy [113].

The role of allogeneic transplantation (allo-HSCT) in this context remains to be defined. Zurko et al., in a multicenter retrospective analysis of 88 patients who underwent allo-HSCT after CAR T-cell therapy, reported a 1-year PFS rate of 45% and an OS rate of 59% [114]. Therefore, in eligible patients, allo-HSCT should be considered for those who achieve a response after CAR T-cell failure. However, limited data exist regarding the efficacy of salvage allo-HSCT following BsAb failure, and the available studies report poor outcomes. Specifically, a retrospective analysis conducted in Spain reported a 2-year OS of approximately 25% following allo-HSCT in patients with previous BsAb failure [115]. Clinical studies are also underway to investigate the first-line use of CAR T-cell therapies (e.g., ZUMA-23) and bispecific antibodies in combination with chemoimmunotherapy. Therefore, given the significant expansion of the therapeutic landscape, determining the optimal sequencing of these treatments will become increasingly important.

In summary, in the context of second-line therapy, ASCT should be considered for patients with chemosensitive DLBCL (e.g., those achieving PR or CR who relapse at least one year after first-line chemoimmunotherapy). Conversely, patients with R/R DLBCL who relapse within one year of first-line treatment should be evaluated for CAR T-cell therapy, given its curative potential of approximately 35–40%. CAR T-cell therapy should also be considered as a second-line option for patients who are ineligible for ASCT due to advanced age or comorbidities. The use of BsAbs in the second-line setting should be considered for selected patients, such as those at high risk of severe CAR T-cell-related toxicities, patients with limited access to CAR T-cell prescribing centers, or those who refuse cellular therapy. However, BsAbs currently represent the best therapeutic option in the third line setting following CAR T-cell therapy failure. Finally, as suggested by the 2025 NCCN guidelines [105], allo-HSCT should be considered for patients achieving a response after progression on CAR T-cell therapy and/or BsAbs.

## 9. Conclusions and Perspectives

The current landscape of treatments in DLBCL is still evolving. Though the introduction of the Pola-R-CHP regimen, after more than a decade, has improved the clinical outcomes of previously untreated DLBCL, 30–45% of patients still experience R/R disease. Over the past 5 years, several chemoimmunotherapy regimens (e.g., polatuzumab vedotin plus BR), chemo-free regimens (e.g., tafasitamab-lenalidomide, bispecific antibodies) and cellular therapies (CAR T-cell) have been introduced for R/R DLBCL. In our opinion, BsAbs represent a game changer in the treatment of lymphoproliferative disorders, owing to their unique mechanism of action, proven efficacy, manageable toxicity profile, greater accessibility and feasibility, compared to other types of therapies (e.g., CAR T-cell, alloSCT). Their incorporation into first-line chemoimmunotherapy or chemo-free regimens may improve cure rates, thereby reducing the proportion of patients who experience R/R disease. Moreover, for those who relapse or are refractory, these agents represent an effective salvage option in this difficult-to-treat population, particularly among patients ineligible for intensive therapies or unable to access costly approaches such as CAR T-cell therapy.

Finally, precision medicine of BsAbs still needs to be developed and expanded. An important tool in this context might be represented by the implementation of liquid biopsy by circulating tumor DNA analysis that may provide a comprehensive molecular characterization of all sites of the disease and may also represent a dynamic tool for disease monitoring during treatment and follow up [116].

The advent of bispecific antibodies has already significantly mutated the treatment paradigms of DLBCL, marking the dawn of a new era in its management.

## Figures and Tables

**Figure 1 cancers-17-03258-f001:**
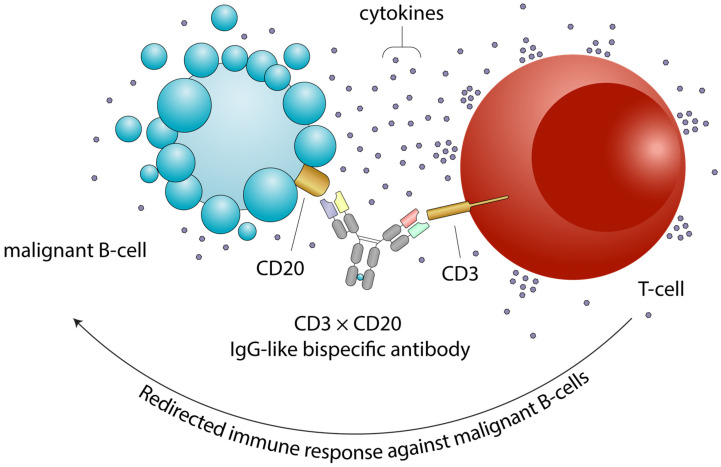
Mechanism of action of bispecific antibodies. Bispecific antibodies simultaneously bind a tumor-associated antigen (e.g., CD20) on the surface of a malignant B-cell and CD3 on the surface of a T cell. This interaction promotes T-cell engagement, activation, and targeted cytotoxicity against the tumor cell. The membrane blebbing and vesicle release depicted on the B-cell surface represent early events of apoptosis following T-cell-mediated killing. Engaged T cells release cytokines and cytotoxic granules that induce apoptosis of the malignant B cell. CRS and ICANS can arise as supraphysiologic responses to immune therapy that activate or engage T-cells and other immune effector cells. Abbreviations: CD, cluster of differentiation; CRS, cytokine release syndrome; ICANS, immune effector cell-associated neurotoxicity syndrome.

**Figure 2 cancers-17-03258-f002:**
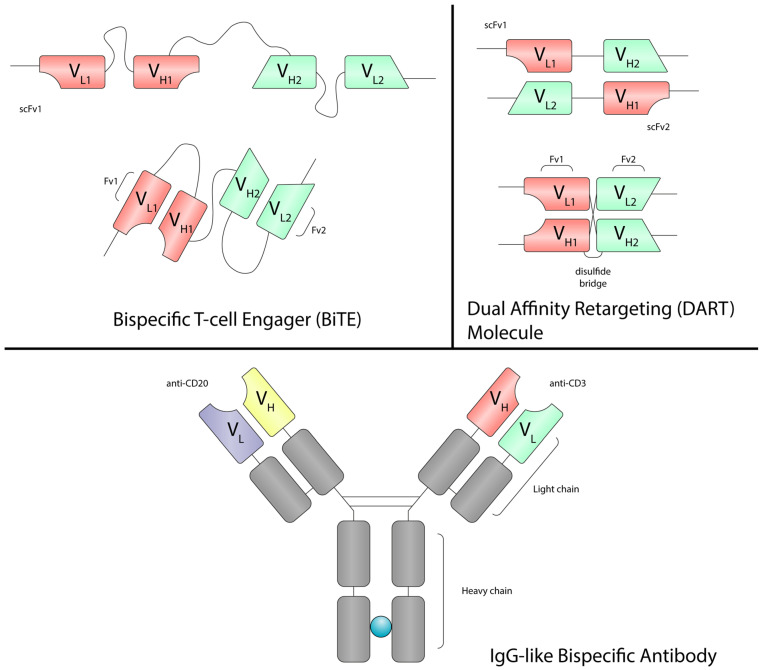
The structure of BiTEs, DARTs and IgG-like bispecific antibodies. BiTEs are bispecific single-chain constructs composed of two scFv linked by a flexible linker and lacking an Fc region. DARTs also consist of two scFv in a stabilized, cross-bridged configuration via disulfide bonds; some incorporate an Fc region (typically IgG1) to enhance half-life and pharmacokinetics. In contrast, IgG-like bispecific antibodies retain the classical IgG structure with two antigen-binding arms and an intact Fc domain. Abbreviations: BiTE, bispecific T-cell engager; DART molecule, dual affinity retargeting molecule; Fv, fragment variant; scFv, single-chain variable fragment; Fc, fragment crystallizable; VH, variable region of the heavy chain; VL, variable region of the light chain; IgG, immunoglobulin G.

**Table 1 cancers-17-03258-t001:** Epcoritamab clinical trials.

Trial Name	NCT	Phase	Population	Experimental Arm	ORR (%)	CR (%)	PFS [Rate/Median]	OS [Rate/Median]	DoR [Rate/Median]	Ref
EPCORE DLBCL-2	NCT05578976	III	Newly dx DLBCL	Epcoritamab + R-CHOP	NA	NA	NA	NA	NA	[38]
EPCORE NHL-2 (Arm 1)	NCT04663347	I/II	Newly dx IPI ≥ 3 DLBCL	Epcoritamab + R-CHOP	100	87	74% at 24 mo/NA	87% at 24 mo/NA	NA	[37]
EPCORE NHL-2 (Arm 8)	NCT04663347	I/II	Newly dx, ineligible for full R-CHOP	Epcoritamab + R-mini-CHOP	89	82	88% at 12 mo/NA	96% at 12 mo/NA	92% at 12 mo/NA	[40]
-	NCT06045247	II	Older/unfit, newly dx DLBCL, ineligible for anthracycline-based therapy	Epcoritamab + R-mini-CVP	NA	NA	NA	NA	NA	-
EPCORE NHL-5 (Arm 3)	NCT05283720	II	Newly dx DLBCL	Epcoritamab + Pola-R-CHP	100	89	NA	NA	NA	[42]
EPCORE DLBCL-3	NCT05660967	II	Newly dx DLBCL unfit/older pts	Epcoritamab monotherapy	74	64	NA/NR	76% at 6 mo/NR	84% at 6 mo/NA	[39]
EPCORE NHL-1	NCT03625037	I/II	R/R LBCL	Epcoritamab monotherapy	63.1	40.1	27.8% at 24 mo/4.4 mo	44.6% at 24 mo/18.5 mo	42.6% at 24 mo/17.3 mo	[36]
EPCORE NHL-5 (Arm 1)	NCT05283720	II	R/R DLBCL	Epcoritamab + lenalidomide	67.6	51.4	NA	NA	NA/NR	[47]
EPCORE DLBCL-1	NCT04628494	III	R/R DLBCL, post-ASCT or ineligible	Epcoritamab	NA	NA	NA	NA	NA	[43]
EPCORE NHL-2 (Arm 4)	NCT04663347	I/II	R/R DLBCL eligible for ASCT	Epcoritamab + R-DHAX/C ± ASCT	76	69	60% at 24 mo/NA	86% at 24 mo/NA	NA	[44]
EPCORE NHL-2 (Arm 5)	NCT04663347	I/II	R/R DLBCL ineligible for ASCT	Epcoritamab + GemOx	85	61	44% at 12 mo/11.2 mo	56.6% at 12 mo/21.6 mo	47.6% at 12 mo/11.7 mo	[45]
EPCORE DLBCL-4	NCT06508658	III	R/R DLBCL, post-ASCT/CAR T-cell ineligible	Epcoritamab + lenalidomide	NA	NA	NA	NA	NA	[46]
EPCORE NHL-5 (Arm 2)	NCT05283720	II	R/R DLBCL	Epcoritamab + lenalidomide + Ibrutinib	NA	NA	NA	NA	NA	-
EPCORE NHL-5 (Arm 4)	NCT05283720	II	R/R DLBCL	Epcoritamab + CC-99282 (CELMoD)	NA	NA	NA	NA	NA	-
-	NCT05852717	II	Transplant/CAR T-cell eligible	Epcoritamab + R-GDP	NA	NA	NA	NA	NA	[48]
-	NCT06287398	II	R/R DLBCL, ASCT eligible	Epcoritamab + R-DHAOx → ASCT + Epcoritamab	NA	NA	NA	NA	NA	-
EpLCART	NCT06414148	II	Post-CAR T-cell responders at high risk	Epcoritamab ± R2	NA	NA	NA	NA	NA	-
-	NCT06458439	II	Pre/post CAR T-cell setting	Epcoritamab before/after CAR T-cell	NA	NA	NA	NA	NA	-

Abbreviations: dx, diagnosis; R/R, relapsed/refractory; IPI, International Prognostic Index; DLBCL, diffuse large B-cell lymphoma; HGBCL, high grade B cell lymphoma; LBCL, large B cell lymphoma; NHL, non-Hodgkin lymphoma; aNHL, aggressive non-Hodgkin lymphoma; R-CHOP, rituximab, cyclophosphamide, doxorubicin, vincristine, prednisone; CHOP, cyclophosphamide, doxorubicin, vincristine, prednisone; M-CHOP, mosunetuzumab, cyclophosphamide, doxorubicin, vincristine, prednisone; Pola-R-CHP, polatuzumab vedotin, rituximab, cyclophosphamide, doxorubicin, prednisone; Pola-M-CHP, polatuzumab vedotin, mosunetuzumab, cyclophosphamide, doxorubicin, prednisone; Pola-CHP, polatuzumab vedotin, cyclophosphamide, doxorubicin, prednisone; R-CVP, rituximab, cyclophosphamide, vincristine, prednisone; R-DHAX/C, rituximab, dexamethasone, high-dose cytarabine, oxaliplatin/carboplatin; R-DHAOx, rituximab, dexamethasone, high-dose cytarabine, oxaliplatin; R-GDP, rituximab, gemcitabine, dexamethasone, cisplatin; GemOx, gemcitabine, oxaliplatin; R-ICE, rituximab, ifosfamide, carboplatin, etoposide; R2, rituximab, lenalidomide; ASCT, autologous stem cell transplantation; axi-cel, axicabtagene ciloleucel; mo, months; NA, not available; NR, not reached. IV, intravenous; SC, subcutaneous.

**Table 2 cancers-17-03258-t002:** Glofitamab clinical trials.

Trial Name	NCT	Phase	Population	Experimental Arm	ORR (%)	CR (%)	PFS [Rate/Median]	OS [Rate/Median]	DoR [Rate/Median]	Ref
NP40126	NCT03467373	I	Newly dx DLBCL	Glofitamab + Pola-R-CHP	100	76.5	NA	NA	NA	[52]
NP40126	NCT03467373	I	Newly dx DLBCL	Glofitamab + R-CHOP	92.9	83.9	NA	NA	NA/NR	[51]
COALITION (Arm B)	NCT04914741	I/II	Newly dx high-risk DLBCL or HGBL	Glofitamab + Pola-R-CHP	98	80	95% at 12 mo	97% at 12 mo	NA	[53]
COALITION (Arm A)	NCT04914741	I/II	Newly dx high-risk DLBCL or HGBL	Glofitamab + R-CHOP	99	70	88% at 12 mo	96% at 12 mo	NA	[53]
-	NCT05800366	II	Newly dx high-risk DLBCL	Glofitamab + Pola-R-CHP	NA	NA	NA	NA	NA	-
SKYGLO	NCT06047080	III	Newly dx LBCL	Glofitamab + Pola-R-CHP	NA	NA	NA	NA	NA	[54]
GO43075	NCT04980222	II	Newly dx high-risk LBCL	Glofitamab + R-CHOP	93.3	80	NA	NA	NA	[55]
GRAIL	NCT06050694	II	Newly dx DLBCL	Pola-R-CHP ± glofitamab	NA	NA	NA	NA	NA	-
-	NCT05798156	II	Older/unfit, newly dx DLBCL	Glofitamab + polatuzumab vedotin	NA	NA	NA	NA	NA	[56]
GLORY	NCT06765317	II	Older/unfit, newly dx DLBCL or HGBCL	Glofitamab + Pola-R-mini-CHP	NA	NA	NA	NA	NA	-
NP30179	NCT03075696	I/II	R/R NHL	Glofitamab monotherapy	52	39	37% at 12 mo/4.9 mo	50% at 12 mo/11.5 mo	64% at 12 mo/18.4 mo	[32]
STARGLO	NCT04408638	III	R/R DLBCL, ASCT ineligible	Glofitamab + GemOx	69.9	57.4	53.2% at 12 mo/14.4 mo	52.8% at 24 mo/25.5 mo	NA/NR	[50]
GO43693	NCT05364424	I	R/R DLBCL, ASCT/CART-cell eligible	Glofitamab + R-ICE	78.1	68.8	NA	NA	NA	[58]
NP39488	NCT03533283	I/II	R/R NHL	Glofitamab + polatuzumab vedotin	80	62	NA/12.3 mo	54% at 24 mo/39.2 mo	NA/21.9 mo	[59]
NP39488	NCT03533283	I/II	R/R NHL	Glofitamab + atezolizumab	36	17	NA	NA	NA	[60]
BP41072	NCT04077723	I/II	R/R NHL	Glofitamab + RO7227166	67.5	56.6	45.6% at 12 mo/9.7 mo	NA/20.9 mo	NA/31.5 mo	[61]
BP43131	NCT05219513	I	R/R NHL	Glofitamab + RO7443904	64	39	NA	NA	NA	[62]
GPL	NCT05335018	II	R/R DLBCL	Glofitamab + poseltinib + lenalidomide	89.3	42.9	55% at 6 mo	81% at 6 mo	66% at 6 mo	[63]
C4971006	NCT05896163	I/II	R/R DLBCL	Glofitamab + maplirpacept	NA	NA	NA	NA	NA	[64]
CO43805	NCT05169515	I	R/R NHL	Glofitamab + CELMoD	NA	NA	NA	NA	NA	-
NCI-2024-00209	NCT06213311	II	R/R LBCL	Glofitamab + axi-cel	NA	NA	NA	NA	NA	[65]
GLOHRT-01	NCT06867536	II	R/R DLBCL	Glofitamab + hypofractionated radiotherapy	NA	NA	NA	NA	NA	-
-	NCT06651853	II	R/R DLBCL	Glofitamab + lenalidomide + hypofractionated radiotherapy	NA	NA	NA	NA	NA	-
-	NCT06570447	II	R/R DLBCL	Glofitamab + tucidinostat	NA	NA	NA	NA	NA	-
CCTG LY18-A	NCT04161248	I	R/R aNHL	Glofitamab + R-GDP	NA	NA	NA	NA	NA	[67]
-	NCT06682130	II	R/R DLBCL	Glofitamab before CAR T-cell	NA	NA	NA	NA	NA	-
UPCC 48420	NCT04889716	II	R/R LBCL	Glofitamab after CAR T-cell	NA	NA	NA	NA	NA	-
LOTIS-7	NCT04970901	I	R/R NHL	loncastuximab tesirine + glofitamab	NA	NA	NA	NA	NA	[66]
BiCAR	NCT04703686	II	R/R NHL after CAR T-cell failure	Glofitamab	76.1%	45.7%	24.5% at 24 mo/3.8 mo	31.2% at 24 mo/14.7 mo	NA/19.7 mo	[68]

Abbreviations: dx, diagnosis; R/R, relapsed/refractory; IPI, International Prognostic Index; DLBCL, diffuse large B-cell lymphoma; HGBCL, high grade B cell lymphoma; LBCL, large B cell lymphoma; NHL, non-Hodgkin lymphoma; aNHL, aggressive non-Hodgkin lymphoma; R-CHOP, rituximab, cyclophosphamide, doxorubicin, vincristine, prednisone; CHOP, cyclophosphamide, doxorubicin, vincristine, prednisone; M-CHOP, mosunetuzumab, cyclophosphamide, doxorubicin, vincristine, prednisone; Pola-R-CHP, polatuzumab vedotin, rituximab, cyclophosphamide, doxorubicin, prednisone; Pola-M-CHP, polatuzumab vedotin, mosunetuzumab, cyclophosphamide, doxorubicin, prednisone; Pola-CHP, polatuzumab vedotin, cyclophosphamide, doxorubicin, prednisone; R-CVP, rituximab, cyclophosphamide, vincristine, prednisone; R-DHAX/C, rituximab, dexamethasone, high-dose cytarabine, oxaliplatin/carboplatin; R-DHAOx, rituximab, dexamethasone, high-dose cytarabine, oxaliplatin; R-GDP, rituximab, gemcitabine, dexamethasone, cisplatin; GemOx, gemcitabine, oxaliplatin; R-ICE, rituximab, ifosfamide, carboplatin, etoposide; R2, rituximab, lenalidomide; ASCT, autologous stem cell transplantation; axi-cel, axicabtagene ciloleucel; mo, months; NA, not available; NR, not reached. IV, intravenous; SC, subcutaneous.

**Table 3 cancers-17-03258-t003:** Odronextamab clinical trials.

Trial Name	NCT	Phase	Population	Experimental Arm	ORR (%)	CR (%)	PFS [Rate/Median]	OS [Rate/Median]	DoR [Rate/Median]	Ref
ELM-1	NCT02290951	I	R/R DLBCL, post-CAR T-cell	Odronextamab monotherapy	48	32	NA/4.8 mo	NA/10.2 mo	NA/14.8 mo	[73]
ELM-2	NCT03888105	II	R/R DLBCL, no prior CAR T-cell	Odronextamab monotherapy	52	31.5	21.1% at 24 mo/4.4 mo	31.6% at 24 mo/9.2 mo	36.9% at 24 mo/10.2 mo	[74,75]
ELM-1/2 pooled	NCT02290951 NCT03888105	I/II	R/R DLBCL	Odronextamab monotherapy	50.8	31.6	17.5% at 36 mo/4.4 mo	27% at 36 mo/9.3 mo	NA/10.5 mo	[69]
OLYMPIA-3	NCT06091865	III	Previously untreated DLBCL	Odronextamab + CHOP	NA	NA	NA	NA	NA	[70]
ATHENA-1	NCT05685173	I	R/R DLBCL	Odronextamab + REGN5837	NA	NA	NA	NA	NA	[71]
CLIO-1	NCT02651662	I	R/R DLBCL	Odronextamab + cemiplimab	NA	NA	NA	NA	NA	[72]
-	NCT06854159	II	R/R DLBCL	Odronextamab pre/post CAR T-cell	NA	NA	NA	NA	NA	-

Abbreviations: dx, diagnosis; R/R, relapsed/refractory; IPI, International Prognostic Index; DLBCL, diffuse large B-cell lymphoma; HGBCL, high grade B cell lymphoma; LBCL, large B cell lymphoma; NHL, non-Hodgkin lymphoma; aNHL, aggressive non-Hodgkin lymphoma; R-CHOP, rituximab, cyclophosphamide, doxorubicin, vincristine, prednisone; CHOP, cyclophosphamide, doxorubicin, vincristine, prednisone; M-CHOP, mosunetuzumab, cyclophosphamide, doxorubicin, vincristine, prednisone; Pola-R-CHP, polatuzumab vedotin, rituximab, cyclophosphamide, doxorubicin, prednisone; Pola-M-CHP, polatuzumab vedotin, mosunetuzumab, cyclophosphamide, doxorubicin, prednisone; Pola-CHP, polatuzumab vedotin, cyclophosphamide, doxorubicin, prednisone; R-CVP, rituximab, cyclophosphamide, vincristine, prednisone; R-DHAX/C, rituximab, dexamethasone, high-dose cytarabine, oxaliplatin/carboplatin; R-DHAOx, rituximab, dexamethasone, high-dose cytarabine, oxaliplatin; R-GDP, rituximab, gemcitabine, dexamethasone, cisplatin; GemOx, gemcitabine, oxaliplatin; R-ICE, rituximab, ifosfamide, carboplatin, etoposide; R2, rituximab, lenalidomide; ASCT, autologous stem cell transplantation; axi-cel, axicabtagene ciloleucel; mo, months; NA, not available; NR, not reached. IV, intravenous; SC, subcutaneous.

**Table 4 cancers-17-03258-t004:** Mosunetuzumab clinical trials.

Trial Name	NCT	Phase	Population	Experimental Arm	ORR (%)	CR (%)	PFS [Rate/Median]	OS [Rate/Median]	DoR [Rate/Median]	Ref
GO29781	NCT02500407	I/II	R/R DLBCL	Mosunetuzumab monotherapy (IV)	42	23.9	22.6% at 12 m/3.2 m	48.1% at 12 m/11.5 m	44.1% at 12 m/7 m	[76]
GO40515	NCT03677141	I/II	Newly dx DLBCL	M-CHOP	87.5	85.0	65.4% at 24 m/NR	NA	66.8% at 24 m/NR	[78]
GO40515	NCT03677141	I/II	Newly dx DLBCL	Pola-M-CHP	75	72.5	70.8% at 24 m/NR	86.3% at 24 m/NR	71.4% at 24 m/NR	[79]
GO40554	NCT03677154	I/II	Previously untreated DLBCL (elderly/unfit)	Mosunetuzumab IV	43	35	39% at 12 m/NA	NA	NA	[80]
GO40554	NCT03677154	I/II	Previously untreated DLBCL (elderly/unfit)	Mosunetuzumab SC + polatuzumab vedotin	55	45	49.7% at 12 m/11.9 m	NA	NA	[81]
-	NCT06828991	II	Newly dx elderly/unfit DLBCL	Mosunetuzumab consolidation post Pola-R-mini-CHP	NA	NA	NA	NA	NA	-
-	NCT06594939	II	Elderly newly dx DLBCL	Mosunetuzumab SC + split-dose Pola-CHP	NA	NA	NA	NA	NA	-
GO40516	NCT03671018	I/II	R/R LBCL	Mosunetuzumab IV + polatuzumab vedotin	46.0	42.9	31.3% at 24 m/11.4 m	48.6% at 24 m/23.3 m	49.7% at 24 m/20.8 m	[82]
GO40516	NCT03671018	I/II	R/R LBCL	Mosunetuzumab SC + polatuzumab vedotin	78	58	64.2% at 12 m/NR	73.8% at 12 m/NR	78.7% at 9 m/NR	[83]
SUMNO	NCT05171647	III	R/R DLBCL	Mosunetuzumab + polatuzumab vedotin	NA	NA	NA	NA	NA	[84]
-	NCT05412290	I	Post-ASCT DLBCL	Mosunetuzumab consolidation	NA	NA	NA	NA	NA	-
-	NCT06015880	I	R/R DLBCL	Mosunetuzumab + polatuzumab vedotin + lenalidomide	NA	NA	NA	NA	NA	-
-	NCT05615636	II	R/R DLBCL	Mosunetuzumab + polatuzumab vedotin + tafasitamab + lenalidomide	NA	NA	NA	NA	NA	-
UPCC 48420	NCT04889716	II	R/R DLBCL	Mosunetuzumab after CAR T-cell	NA	NA	NA	NA	NA	-
-	NCT05260957	II	R/R DLBCL	CAR T-cell + mosunetuzumab + polatuzumab vedotin	NA	NA	NA	NA	NA	-
-	NCT05315713	I/II	R/R DLBCL	Mosunetuzumab + tiragolumab ± atezolizumab	NA	NA	NA	NA	NA	-
CO43805	NCT05169515	I	R/R DLBCL	Mosunetuzumab + CELMoD	NA	NA	NA	NA	NA	-
LOTIS-7	NCT04970901	I	R/R NHL	Loncastuximab tesirine + mosunetuzumab	NA	NA	NA	NA	NA	[66]
-	NCT05672251	II	R/R DLBCL	Loncastuximab tesirine + mosunetuzumab	NA	NA	NA	NA	NA	[85]

Abbreviations: dx, diagnosis; R/R, relapsed/refractory; IPI, International Prognostic Index; DLBCL, diffuse large B-cell lymphoma; HGBCL, high grade B cell lymphoma; LBCL, large B cell lymphoma; NHL, non-Hodgkin lymphoma; aNHL, aggressive non-Hodgkin lymphoma; R-CHOP, rituximab, cyclophosphamide, doxorubicin, vincristine, prednisone; CHOP, cyclophosphamide, doxorubicin, vincristine, prednisone; M-CHOP, mosunetuzumab, cyclophosphamide, doxorubicin, vincristine, prednisone; Pola-R-CHP, polatuzumab vedotin, rituximab, cyclophosphamide, doxorubicin, prednisone; Pola-M-CHP, polatuzumab vedotin, mosunetuzumab, cyclophosphamide, doxorubicin, prednisone; Pola-CHP, polatuzumab vedotin, cyclophosphamide, doxorubicin, prednisone; R-CVP, rituximab, cyclophosphamide, vincristine, prednisone; R-DHAX/C, rituximab, dexamethasone, high-dose cytarabine, oxaliplatin/carboplatin; R-DHAOx, rituximab, dexamethasone, high-dose cytarabine, oxaliplatin; R-GDP, rituximab, gemcitabine, dexamethasone, cisplatin; GemOx, gemcitabine, oxaliplatin; R-ICE, rituximab, ifosfamide, carboplatin, etoposide; R2, rituximab, lenalidomide; ASCT, autologous stem cell transplantation; axi-cel, axicabtagene ciloleucel; mo, months; NA, not available; NR, not reached. IV, intravenous; SC, subcutaneous.

**Table 5 cancers-17-03258-t005:** Summary of key details of the discussed bispecific antibodies.

Bispecific Antibody	Route of Administration	Duration of Therapy	Currently Approved Indication	ORR [CR] (%) *	CRS (%)	ICANS (%)
Epcoritamab	SC	UP	R/R DLBCL	63.1 [40.1]	51	6.4
Glofitamab	IV	FD	R/R DLBCL ^†^	52 [39]	63	8
Odronextamab	IV	UP	R/R FL R/R DLBCL	50.8 [31.6] ^‡^	52.9	0
Mosunetuzumab	IV	FD	R/R FL	42% [23.9]	26.1	1.1

Abbreviations: CRS, cytokine release syndrome; ICANS, immune effector cell-associated neurotoxicity syndrome; SC, subcutaneous; IV, intravenous; UP, until progression; FD, fixed-duration; R/R, relapsed/refractory; DLBCL, diffuse large B-cell lymphoma; FL, follicular lymphoma; GemOx, gemcitabine and oxaliplatin. † glofitamab as monotherapy is indicated for patients with R/R DLBCL after the second line of therapy; glofitamab in combination with GemOx is indicated for the treatment of patients with R/R DLBCL who are ineligible for ASCT. * the ORR and CR rates are reported for a follow-up of ~25 months for epcoritamab, ~36 months for glofitamab, a median of 23 months for odronextamab, and a median of 10.1 months for mosunetuzumab. ‡ the ORR and CR rates reported refer to pooled analysis of ELM-1 and ELM-2 trials.

**Table 6 cancers-17-03258-t006:** CRS and ICANS AEs according to bispecific antibodies.

Bispecific Antibody	CRS (%)	CRS Low-Grade (1–2) (%)	Typical CRS Onset	Median Time to CRS Onset (h)	Median Time to CRS Resolution (h)	ICANS (%)
Epcoritamab	51	47.8	Cycle 1	20	48	6.4
Glofitamab	63	59.0	Cycle 1	13.5	30.5	8
Odronextamab	52.9	48	Cycle 1	17.8	7.1	0
Mosunetuzumab	26.1	23.9	Cycle 1	NA	26	1.1

Abbreviations: CRS, cytokine release syndrome; ICANS, immune effector cell-associated neurotoxicity syndrome; AEs, adverse events; NA, not available.

## Data Availability

No new data were created or analyzed in this study. Data sharing is not applicable to this article.

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
