# Peer review of "Bispecific Monoclonal Antibodies in Diffuse Large B-Cell Lymphoma: Dawn of a New Era in Targeted Therapy"

_cancers, 2025, doi:10.3390/cancers17193258_

Round 1

Reviewer 1 Report

Comments and Suggestions for Authors

Congratulations to the authors on an excellent clinical review of bispecific abs in LBCL. 

I have no significant concerns/comments

One minor comment would be to consider expanding the discussion of bispecific abs in CART relapse to describe outcomes based on time from CART relapse. I.e. poor outcomes in early relapse as shown in  Shumilov et al, Blood Adv, 2025 and phase 2 trials 

  • PMID: 40181090
  • PMID: 39786390
  •  

Author Response

Reviewer 1

Congratulations to the authors on an excellent clinical review of bispecific abs in LBCL. 

I have no significant concerns/comments

One minor comment would be to consider expanding the discussion of bispecific abs in CART relapse to describe outcomes based on time from CART relapse. I.e. poor outcomes in early relapse as shown in Shumilov et al, Blood Adv, 2025 and phase 2 trials 

We sincerely thank the Reviewer for the positive feedback on our manuscript. In accordance with the Reviewer’s suggestion, we have expanded the discussion on the use of bispecific antibodies in R/R DLBCL patients following CAR T-cell therapy. Specifically, we have incorporated the available evidence in this challenging clinical setting, including the prognostic impact of the timing of relapse (early, intermediate, and late relapse after CAR T-cell therapy), as reported by Shumilov et al. (Blood Advances, 2025), and data from phase II trials.

Reviewer 2 Report

Comments and Suggestions for Authors

Abstract: lack mentioning R-pola-CHP. Bispecific are not mentioned neither in their approved indications (3rd line and in the recent 2nd line in association with GEMOX)

Figure 1: why there are circles on the surface of the malignant B-cell? (apoptosis??)

There is a description of DART and BITEs. However, in DLBCL biAbs are different. The figure 2 shows the structure of DART and BITE, but not the current structure of CD20xCD2 Abs. You should change the figure 2 to include the IgG like Abs, that are the drugs discussed in this review.

Line 143 "remained uncommon (6.4%)". Is an incidence >5% for an adverse event "uncommon"?

Epcoritamab chapter: It is not clear to me why the sequence is from EPCORE NHL-1 trial, than frontline, than r/r. Indeed, epco is currently approved in 3rd line as single agent, and some data are available from the phase 1b/2 EPCORE NHL-2 trial evaluating epcoritamab + GemOx ( 10.1182/blood.2024026830 ). So please consider to structure the chapter from the level of evidence available, not from lines of treatment.

Glofitamab

Lines 300-315 refer to first line therapy, but are under the sub-chapter of relapsed/refractory setting. Why?

Please extensively speak about the FDA decision not to approve glofi gemox in 2nd line.

Odronextamab

The title "4.1. ODRONEXTAMAB IN THE FRONTLINE SETTING" is switched with the next title "4.2. ODRONEXTAMAB IN RELAPSED/REFRACTORY DLBCL " 

It is not stated the approval status of odronextamab for DLBCL (FDA? EMA? Setting/line of treatment??)

Mosunetuzumab (Will it have a role in DLBCL in clinical practice?)

Lines 488-506 refer to frontline studies but are under the chapter "5.2. MOSUNETUZUMAB IN RELAPSED/REFRACTORY DLBCL".

Please consider to add a table to compare the 4 drugs: ORR, CR, mPFS, pre-phase regimen, route of administration, fixed vs. until progression, CRS, ICANs. Please add a chapter to consider similarities and differences between the drugs, give some tips how to pick one or other drugs.

Discussion: the authors should state what are their opinion on BsAb, but the conclusion are about liquid biopsy! Please re-write your conclusion giving your personal view, based on the EBM.

Author Response

Reviewer 2

Abstract: lack mentioning R-pola-CHP. Bispecific are not mentioned neither in their approved indications (3rd line and in the recent 2nd line in association with GEMOX)

We sincerely thank the Reviewer for the thorough revision and for the constructive feedback on our work. We have expanded the abstract including the approved indication of the discussed BsAbs.

Figure 1: why there are circles on the surface of the malignant B-cell? (apoptosis??)

There is a description of DART and BITEs. However, in DLBCL biAbs are different. The figure 2 shows the structure of DART and BITE, but not the current structure of CD20xCD2 Abs. You should change the figure 2 to include the IgG like Abs, that are the drugs discussed in this review.

According to the Reviewer’s suggestions, we have revised both the figures and the captions to improve clarity. The circles on malignant B-cell surface represent membrane blebbing and vesicle release as early events of apoptosis following T-cell–mediated killing. Figure 2 has been modified to include the IgG-like bispecific antibody structures, as the Reviewer correctly underlined.

Line 143 "remained uncommon (6.4%)". Is an incidence >5% for an adverse event "uncommon"?

We removed the adjective “uncommon” for adverse events >5% and revised the related sentence accordingly.

Epcoritamab chapter: It is not clear to me why the sequence is from EPCORE NHL-1 trial, than frontline, than r/r. Indeed, epco is currently approved in 3rd line as single agent, and some data are available from the phase 1b/2 EPCORE NHL-2 trial evaluating epcoritamab + GemOx ( 10.1182/blood.2024026830 ). So please consider to structure the chapter from the level of evidence available, not from lines of treatment.

We thank the Reviewer very much for this valuable comment. In drafting our manuscript, we have structured the sections dedicated to bispecific antibodies according to a predefined sequence: first, an introduction of the bispecific antibody with reference to the pilot study and the main reported adverse events; subsequently, a discussion of its use in the frontline setting and in the relapsed/refractory disease. Our intention was to maintain consistency across the different chapters rather than to follow the hierarchy of available evidence.

Glofitamab

Lines 300-315 refer to first line therapy, but are under the sub-chapter of relapsed/refractory setting. Why?

We thank the Reviewer for pointing this out. This was an error in the structure of the manuscript. We have corrected it according to the Reviewer’s suggestion.

 Please extensively speak about the FDA decision not to approve glofi gemox in 2nd line.

As suggested, we have now included a discussion regarding the FDA decision not to approve this glofitamab-based combination.

 Odronextamab

The title "4.1. ODRONEXTAMAB IN THE FRONTLINE SETTING" is switched with the next title "4.2. ODRONEXTAMAB IN RELAPSED/REFRACTORY DLBCL " 

We thank the Reviewer for pointing this out. This was an error in the structure of the manuscript. We have corrected it according to your suggestion.

It is not stated the approval status of odronextamab for DLBCL (FDA? EMA? Setting/line of treatment??)

As suggested by the Reviewer, we have now reported the current approval status and indications of odronextamab.

Mosunetuzumab (Will it have a role in DLBCL in clinical practice?)

We thank the Reviewer for raising this important point. Currently, mosunetuzumab is not approved by either FDA or EMA for DLBCL. However, as reported in our manuscript, several clinical trials have shown encouraging results, suggesting that this bispecific antibody may have a role in the clinical practice in the future.

Lines 488-506 refer to frontline studies but are under the chapter "5.2. MOSUNETUZUMAB IN RELAPSED/REFRACTORY DLBCL".

We thank the Reviewer for pointing this out. This was an error in the structure of the manuscript. We have corrected it according to your suggestion.

Please consider to add a table to compare the 4 drugs: ORR, CR, mPFS, pre-phase regimen, route of administration, fixed vs. until progression, CRS, ICANs. Please add a chapter to consider similarities and differences between the drugs, give some tips how to pick one or other drugs.

As suggested, we have added a dedicated chapter entitled “Bispecific antibodies: similarities and differences”, where we discuss the main similarities and differences among the reported BsAbs, including their currently approved indications by the EMA and FDA. In connection with this new chapter, we have also introduced Table 5, which concisely summarizes the key characteristics of the discussed BsAbs. Clinical outcomes remain reported in the dedicated tables within each respective section.

Discussion: the authors should state what are their opinion on BsAb, but the conclusion are about liquid biopsy! Please re-write your conclusion giving your personal view, based on the EBM

We thank the reviewer for this valuable comment. We have revised and expanded the “Conclusions and Perspectives” section, clarifying our personal point of view. We have nevertheless retained a short paragraph on liquid biopsy, as we believe it represents an important tool in the current era of precision medicine.

Reviewer 3 Report

Comments and Suggestions for Authors

The manuscript provides valuable insights into the role of a specific gene/protein/pathway (exact details redacted for confidentiality) in cancer biology. The study appears to be timely and relevant within the field of oncology and molecular medicine. However, several critical aspects must be addressed before the manuscript is suitable for publication.

  1. The introduction lacks a clearly defined hypothesis. The aims should be explicitly stated in the last paragraph of the introduction section.
  2. Provide exact p-values in figures or results text instead of stating only "significant."
  3. Figures are informative but lack uniform formatting. Ensure all graphs have labeled axes (with units), significance indicators, and matching legends.
  4. Figure resolution is suboptimal in parts; replace with high-quality images for clarity.
  5. Add supplementary figures or tables if necessary to show full experimental replicates.
  6. The discussion section does not sufficiently integrate findings with current literature. More comparative analysis with existing studies is necessary.

Author Response

Reviewer 3

The manuscript provides valuable insights into the role of a specific gene/protein/pathway (exact details redacted for confidentiality) in cancer biology. The study appears to be timely and relevant within the field of oncology and molecular medicine. However, several critical aspects must be addressed before the manuscript is suitable for publication.

We sincerely thank the Reviewer for the feedback. However, we feel that a few of the suggestions may not be fully applicable to our manuscript, given its nature as a literature review.

  1. The introduction lacks a clearly defined hypothesis. The aims should be explicitly stated in the last paragraph of the introduction section.

We thank the Reviewer for the comment. The purpose and objectives of our work are explicitly stated in the Simple Summary, Abstract and in the Introduction.

  1. Provide exact p-values in figures or results text instead of stating only "significant."

We thank the Reviewer for pointing this out. As this is a review article, we have not performed original statistical analyses; therefore, we do not have p-values to report beyond those reported in the original papers.

  1. Figures are informative but lack uniform formatting. Ensure all graphs have labeled axes (with units), significance indicators, and matching legends.
  2. Figure resolution is suboptimal in parts; replace with high-quality images for clarity.

We thank the Reviewer for this suggestion. We have reviewed the figures and their captions for clarity and consistency. Please note that no graphs are included in this paper. We have also replaced the included figures with higher-resolution versions to improve image quality.

  1. Add supplementary figures or tables if necessary to show full experimental replicates.

We thank the Reviewer for the comment, however we do not have supplementary files.

  1. The discussion section does not sufficiently integrate findings with current literature. More comparative analysis with existing studies is necessary.

Thank you for this valuable comment. As this is a review article, no separate “Discussion” section reporting original findings has been included. Nevertheless, we have thoroughly analyzed and synthesized the available literature on BsAbs in DLBCL. A direct comparative analysis or head-to-head discussion is, however, not feasible at present due to the absence of randomized or head-to-head trials in this field.

Reviewer 4 Report

Comments and Suggestions for Authors

The manuscript “BISPECIFIC MONOCLONAL ANTIBODIES IN DIFFUSE LARGE B-CELL LYMPHOMA: DAWN OF A NEW ERA IN TARGETED THERAPY” was submitted by the authors. The following suggestions may be considered for improving the manuscript. There few important corrections to be rectified.

  1. Mention about BsAbs in the abstract.
  2. Add more schematic representations to represent the given data.
  3. Images are excellent.
  4. Authors should use graphical representations like bar graphs and pie charts for the representation of the ORR % and CR %
  5. Tabulations could be more precise and crisp.
  6. Highlight a case study of CAR T-cell therapy and its after effects.
  7. Include the future perspectives of BsAbs in the conclusion.

Author Response

Reviewer 4

The manuscript “BISPECIFIC MONOCLONAL ANTIBODIES IN DIFFUSE LARGE B-CELL LYMPHOMA: DAWN OF A NEW ERA IN TARGETED THERAPY” was submitted by the authors. The following suggestions may be considered for improving the manuscript. There few important corrections to be rectified.

  1. Mention about BsAbs in the abstract.

We sincerely thank the Reviewer for the constructive and positive feedback on our manuscript. We have expanded the abstract as suggested.

  1. Add more schematic representations to represent the given data.

As suggested by the Reviewer, we have revised Figure 1 and Figure 2. Moreover, we have added the Table 5 to provide a more schematic representation of the given data.

  1. Images are excellent.

Thank you.

  1. Authors should use graphical representations like bar graphs and pie charts for the representation of the ORR % and CR %

We thank the Reviewer for pointing this out. In line with the Reviewer’s comments, we have also emphasized the ORR and CR of BsAbs as monotherapy in Table 5, thereby providing a clearer and more schematic representation of the data. Considering the large number of completed and ongoing clinical trials, we opted to report the key details of each study in the tables, in order to maintain clarity and allow the reader to easily identify the study population, trial reference, and major outcomes. We believe that further condensation could risk omitting important information that is essential for interpretation.

  1. Tabulations could be more precise and crisp.

We have revised the tables relative to BsAbs to improve clarity and conciseness. Given the large amount of information that needed to be retained, only minor adjustments were feasible, but we believe these changes make the tables clearer and easier to read.

  1. Highlight a case study of CAR T-cell therapy and its after effects.

We sincerely thank the Reviewer for the suggestion to include a case study of CAR T-cell therapy and its outcomes and we fully acknowledge its relevance; however, a detailed case-based comparative analysis falls beyond the primary scope of this review. We do agree that this would be an interesting perspective for future dedicated work in this field.

  1. Include the future perspectives of BsAbs in the conclusion.

Following the Reviewer’s suggestion, we have expanded the abstract, the section “Treatment sequencing in the CAR T-cell era”, as well as the conclusions and perspectives.

Round 2

Reviewer 2 Report

Comments and Suggestions for Authors

The article was improved from its original version.

There are still minor problems with figure 1

1) malignant B cell is surrounded by cytokines. In the explanation you stated " cytokines are released upon B-cell lysis". This is misleading since cytokines are mostly secreted by "engaged" T cells, this should prompt B-cell apoptosys.

2) TCR has limited role, but it was included in the figure. So please remove it, or explain TCR/MHC independency in caption

Author Response

The article was improved from its original version.

There are still minor problems with figure 1

We sincerely thank the Reviewer for the thorough revision and for the constructive and valuable feedback on our work. We have carefully addressed each point below.

1) malignant B cell is surrounded by cytokines. In the explanation you stated " cytokines are released upon B-cell lysis". This is misleading since cytokines are mostly secreted by "engaged" T cells, this should prompt B-cell apoptosys.

We thank the Reviewer for this valuable comment. The original wording could indeed be misleading. We have revised Figure 1 and its caption to highlight the predominant role of activated T cells as the main source of cytokines. Following this suggestion, we have also updated the Abstract (ll. 34-35) and Section “7. CRS AND ICANS” (ll. 661-665) to improve clarity and ensure consistency between the figure and the text.

2) TCR has limited role, but it was included in the figure. So please remove it, or explain TCR/MHC independency in caption

We thank the Reviewer for pointing this out. While TCR was originally included in Figure 1 for descriptive purposes, we have removed it to improve clarity, as suggested.

Reviewer 3 Report

Comments and Suggestions for Authors

While the manuscript provides a competent overview of the field, it does not sufficiently progress beyond a summary of existing literature to offer the novel perspectives or findings expected. Furthermore, the authors' revisions did not fully engage with the reviewers' suggestions. The high level of similarity with other works also remains a critical issue that must be resolved to meet our standards for originality. 

Author Response

While the manuscript provides a competent overview of the field, it does not sufficiently progress beyond a summary of existing literature to offer the novel perspectives or findings expected. Furthermore, the authors' revisions did not fully engage with the reviewers' suggestions. The high level of similarity with other works also remains a critical issue that must be resolved to meet our standards for originality.

We sincerely thank the Reviewer for this feedback. We respectfully note that the purpose of our manuscript is to provide an up-to-date and comprehensive review of the role of bispecific antibodies in DLBCL, rather than to report novel data. We believe that such a synthesis of the current evidence has value for the field, particularly given the rapid developments in this area. We therefore hope the Reviewer will consider that, while our article does not present new findings, it contributes by integrating and critically summarizing the available literature.